# Prevalence and associated factors of pregnancy induced hypertension among pregnant women in public hospitals of Hadiya Zone, Central Ethiopia: A cross-sectional study

Zerfework Debebe Argago[1,2]*, Nebiyu Dereje[3], Neena Elezebeth Philip[1]

1 Department of Epidemiology and Public Health, School of Life Science, Central University of Tamil Nadu, Thiruvarur, India, 2 Department of Midwifery, College of Medicine and Health Sciences, Wachemo University, Hossana, Ethiopia, 3 School of Public Health, Wachemo University, Hossana, Ethiopia

* zerfedebebe@gmail.com

## Abstract

### Background

Pregnancy-induced hypertension is one of the global public health burdens contributing to several adverse perinatal outcomes. However, data on the prevalence and associated factors of PIH is limited in Hadiya Zone, central Ethiopia.

### Methods

We conducted a cross-sectional study among women attending antenatal care in the four public hospitals in the Hadiya zone from August 1, 2023, to January 30, 2024. The total sample size (433) was proportionally allocated to each hospital. Data were collected by face-to-face interview using a structured questionnaire. Blood pressure was measured by a digital Sphygmomanometer, and Pregnancy-induced hypertension status was considered if systolic blood pressure was ≥ 140 mmHg and/or diastolic blood pressure ≥90 mmHg on two measurements among pregnant women with gestational age > 20 weeks. Factors associated with Pregnancy-induced hypertension were identified by multivariable binary logistic regression analysis, as expressed by adjusted odds ratio (aOR) and its 95% confidence interval.

### Results

The mean age of the participants was 29.4 years (SD = ±4.6), and the majority of women were multiparous (59.6%). The prevalence of Pregnancy-induced hypertension was 13.4% (95% CI: 10.2% – 16.9%), of which 81% were preeclampsia. In the multivariable analysis, PIH was associated with, the poorest wealth index (aOR = 0.26, 95%CI: 0.08–0.90), having a family history of hypertension (aOR =

**Data availability statement:** All relevant data are within the paper.

**Funding:** This study was supported by a grant from Wachemo University. The funder had no role in study design, data collection and analysis, decision to publish, or preparation of the manuscript.

**Competing interests:** The authors have declared that no competing interests exist.

12.52,95%CI: 4.52–34.62), overweight maternal BMI (aOR = 4.24, 95%CI: 2.06–8.72), and gestational age (aOR = 0.21 at 95% CI: 0.06–0.82).

## Conclusions

More than 13 out of 100 pregnant women were found to have pregnancy-induced hypertension in the Hadiya zone, Southern Ethiopia. The high prevalence of Pregnancy-induced hypertension and its association with wealth index, gestational age, overweight/obesity and family history of hypertension/Pregnancy-induced hypertension underscores the need for targeted, tailored, and contextualized interventions to address Pregnancy-induced hypertension in the Hadiya zone. Creating awareness on Pregnancy-induced hypertension among pregnant women and the general population and enhancing early screening and detection services in the community and healthcare facilities need to be strengthened in the Hadiya zone.

## Introduction

Pregnancy-induced hypertension (PIH) is one of the global public health burdens contributing to several adverse perinatal outcomes [1]. Pregnancy-induced hypertension (PIH) is defined as hypertension that occurs after 20 weeks of gestation in women with previously normal blood pressure [2]. Pregnancy-induced hypertension is further classified as gestational hypertension, preeclampsia, and eclampsia [2]. When the systolic blood pressure is ≥ 160 mmHg, or the diastolic blood pressure is ≥ 110 mmHg based on the average of at least two measurements taken at least 15 minutes apart using the same arm, it is considered severe preeclampsia. If seizure accompanies pregnancy-induced hypertension, it is considered eclampsia [3–5].

PIH is the third leading cause of fetal and maternal death. Due to pregnancy-related causes, around 287,000mothers died globally in 2020, out of which 14% accounted for pregnancy-induced hypertension [6,7].Only 2–8% of pregnancies are complicated with PIH in high-income countries, whereas, in low-income countries, the magnitude of PIH reaches up to 16.7%. For instance, in low-income countries, the prevalence of pregnancy-induced hypertension was notably higher in some countries. In Pakistan, 9.3%, in Mozambique, 10.9%, and in Nigeria, 10.2%, and in Bangladesh 14.4%.Conversely, in high-income countries, the prevalence of PIH was significantly lower, with Japan at 4.6% and Sweden at 4.4% [8].

PIH has several maternal and fetal adverse effects. It increases the risk of death, heart attack, cardiac failure, cerebral vascular accidents, stroke, disseminated intravascular coagulation, and multiple organ failure in a woman. It increases the risk of prenatal complications to the fetus, such as poor placental transfer of oxygen, fetal growth restriction, preterm birth, placental abruption, stillbirth, and neonatal death [4,9]. A prior study conducted in Ethiopia revealed that prenatal mortality linked to PIH is one of the highest figures in the world, with 111 deaths/1000 live births [10]. According to a hospital-based retrospective study in Ethiopia, 35% of maternal deaths were associated with PIH [10].

The cause of hypertension is still obscure, but there are certain risk factors accepted to influence the chance of developing pregnancy-induced hypertension. These risk components include different pregnancies, nulliparity, a history of chronic hypertension, gestational diabetes, obesity, fetal mutation, extraordinary maternal age (under 20 or over 40 years), a history of PIH in previous pregnancies, chronic illnesses such as renal infection, diabetes mellitus, and cardiac disease, unrecognized chronic hypertension, a positive family history of PIH indicating hereditary susceptibility, psychological stress, alcohol use, rheumatic arthritis, extraordinary underweight or overweight, asthma, and low socioeconomic status [2,11].

The prevalence of PIH in Ethiopia varies from 2.2% to 18.3% [2,12–14]. Although they are inconclusive and inconsistent, studies in Ethiopia reported that family history of pregnancy induced-hypertension, kidney disease, gestational age, rural residents, educational status, family history of hypertension, maternal age, diabetes mellitus, prim gravid, multiparity were associated with PIH [2,6,11,15,16].

PIH represents a significant public health concern that necessitates focus because of its effects on expectant mothers and their babies. In Ethiopia, maternal mortality remains a significant public health issue. In 2020, there were approximately 267 maternal deaths per 100,000 live births, and pregnancy-induced hypertension plays a significant role in maternal mortality [6]. A review study on the causes of maternal mortality in Ethiopia revealed that the proportion of maternal deaths attributed to hypertensive disorders increased from 4% to 29% between 1980 and 2012 [17]. The Federal Ministry of Health has implemented several strategies to decrease the morbidity and mortality rates of both mothers and newborns, these efforts include enhancing access to and bolstering facility-based maternal and newborn services. However, it is concerning that the prevalence of maternal morbidity and mortality caused by pregnancy induced hypertension is on the rise [17].

Although pregnancy-induced hypertension is a significant contributor to maternal morbidity and mortality, there is limited understanding of its current prevalence and associated factors among women receiving antenatal care in Ethiopia, particularly in the Hadiya zone. As Ethiopia is diverse in terms of cultural practices, socio-economic status [e.g., wealth index], and access to healthcare facilities, it is critical to understand contextual factors that affect PIH. Therefore, this study aimed to assess the prevalence of PIH and its associated factors among women attending antenatal care services in public hospitals of the Hadiya zone, Southern Ethiopia.

## Materials and methods

### Study design and settings

A facility-based, quantitative, cross-sectional study was conducted in Hadiya Zone public hospitals in Ethiopia from August 1, 2023, to January 30, 2024. Hadiya Zone is one of 14 zones located in the Central Ethiopia Regional State—about 233 km from Addis Ababa. A census was conducted in 2015; the Hadiya zone has a total of 1,231,196, of whom 612,026 are men and 612,170 women, with an area of 3,593.31 square kilometers and a population density of 342.64. At the same time, 134,041 or 208.599 or 89.11% are rural inhabitants. A total of 231,846 households were counted. The largest reported ethnic group in this zone is Hadiya 75.35% of the population said they were protestant religious followers, 11.13% were Muslim, 8.45% were Ethiopian Orthodox Christianity, and 4.31% were catholic. The zone has 541 health facilities that offer maternity services. Four hospitals include one teaching hospital, three primary hospitals, 61 health centres, more than 162 private clinics, and 311 health posts. On average, the total number of mothers serviced by all hospitals in 2021 was 10,774. This is based on the number of annual registrations at each hospital [18].

### Source and study population

The source population was all pregnant women who attended ANC service in the Hadiya zone public hospitals from August 1, 2023, to January 30, 2024. The study population was pregnant women who visited ANC follow-up in the selected hospitals of the Hadiya zone, whose gestational age was greater than 20 weeks during the study period and who consented to be included in this study.

### Inclusion and exclusion criteria

Mothers whose gestational age is above 20 weeks and attending ANC services were included in the study, whereas mothers who have a previous history of hypertension and are severely ill or unable to respond to the interview questions were excluded from the study.

### Sample size and sampling technique

**Sample size determination.** To assess the prevalence of PIH and its associated factors, the sample size was calculated using the single population proportion formula by considering 95% CI, the prevalence of pregnancy-induced hypertension in SNNPR Ethiopia 10.3% [9], and 3% margin of error, and 10% non-response rate. Accordingly, the calculated sample size is 433 samples.

**Sampling procedure.** The study was conducted in all the public hospitals located in the Hadiya zone. Four public hospitals in the zone are Nigist Elleni Mohammed Memorial Comprehensive Specialized Hospital (NEMMCSH), Shone Primary Hospital, Gimbichu Primary Hospital, and Homecho Primary Hospital. To assess the prevalence of PIH and its associated factors, the estimated sample size (n = 433) was allocated proportionally to the hospitals, as described in Fig 1 below. The total number of expected pregnant women, with GA > 20 weeks, at each hospital was obtained from the respective hospitals based on the number of pregnant women who attended in the last year of the same period. Pregnant women attending ANC in the respective hospital were included consecutively until the desired sample size was reached (Fig 1).

### Data collection tools and procedures

Data was collected by using an administered structured questionnaire, which is adapted from relevant literature [19]. The questionnaire was prepared in English, translated into Amharic, and then translated back into English to check for consistency by independent translators. Five midwives and two supervisors were involved in the data collection process. Data on proteinuria and other clinical variables was taken from the medical charts of pregnant women. Measurement of blood

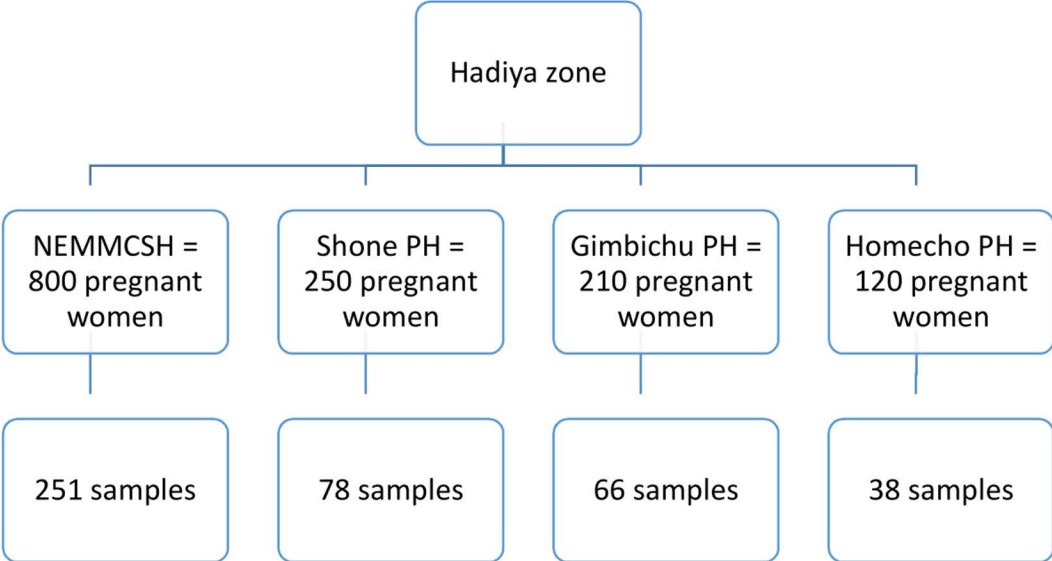

**Fig 1. Schematic diagram showing the sampling distribution of women attending antenatal care at public Health Hospitals, Hadiya zone, Ethiopia, August 1, 2023, to January 30, 2024.**

pressure was taken by using standard procedures, with a digital sphygmomanometer; the woman was instructed to sit quietly and rest for 15 minutes with their legs uncrossed. The measurements were taken with the participant sitting upright with proper back support and with the right arm supported on a table or a surface at the level of the heart. The cuff was inflated at 2–3 mmHg per second. Systolic blood pressure (SBP) was taken up on hearing the first sound, and diastolic blood pressure (DBP) was taken up on the fourth (muffled) Korotkoff sound. Three readings were taken at five-minute intervals, and the average was used as the final blood pressure measurement. Anthropometric data (weight and height) was taken using standard procedures.

Data collectors were trained for three days on questions included within the questionnaire, interviewing methods, the purpose of the study, and the significance of privacy, confidentiality, discipline, and approach to the interviewees and the responses. Before conducting the main study, a pre-test was carried out in 5% of the sample, and appropriate corrections were made to the questionnaire based on the pre-test findings.

### Data processing and analysis

Data was entered into Epi info, cleaned and checked for consistency and completeness, and exported into SPSS version 25.0 software for analysis. Descriptive statistics were used to summarize the findings by examining the frequency and proportions of categorical variables. The mean/median and standard deviation/interquartile ranges were reported for numerical variables.

The prevalence of pregnancy-induced hypertension (PIH) was determined by the proportion of those pregnant women with hypertension out of all pregnant women included in the study and expressed by its respective 95% confidence interval (CI). Factors associated with PIH were identified using multivariable binary logistic regression analysis. Those variables with P value ≤0.25 in the bivariable analysis and those with clinical relevance were used as candidate variables for the multivariable analysis. The measure of the association was estimated by the adjusted odds ratio (AOR) and its respective 95% CI, and the significance level was set at 5%. The goodness of fit of the model was determined by the Hosmer-Lemeshow's test.

### Study variables

**Dependent variable.** Pregnancy-Induced hypertension (1=Yes,0=No).

**Independent variables. Socio-demographic variables:** Maternal age, residence, educational status, occupation and wealth index.

**Obstetrics variables:** Parity, gestational age at interview, number of ANC follow up, Mode of delivery.

**Medical variables:** Previous history of pregnancy-induced HTN, History of D/M, anaemia and renal disease.

**Personal behavioural variable:** History of alcohol use, smoking and chat chewing.

### Operational definitions

**Pregnancy-induced hypertension (PIH)** "is systolic blood pressure ≥ 140 mm Hg or diastolic blood pressure ≥ 90 mmHg after gestation age of twenty weeks in women with previously normal blood pressure".

**Preeclampsia** "is systolic blood pressure ≥ 140 mm Hg or diastolic pressure ≥ 90 mm Hg, which occurs at twenty weeks of gestational age onwards on a previously normal hypertensive woman with proteinuria of 1+ or 2+ on the dipstick".

**Body mass index:** To measure Body Mass Index (BMI): First, we gathered participants' weight in kilograms (kg) and their height in meters (m). Then, use the formula **BMI = weight (kg)/ (height (m)) ²** for metric units; this calculation will help us assess their body weight relative to their height. BMI was categorized according to the 2009 IOM classification: underweight (BMI < 18.5), normal weight (BMI 18.5–24.9), overweight (BMI 25.0–29.9), and obese (BMI ≥ 30.0).

                                                                                         

**Hypertensive disorders during pregnancy:** "This includes Chronic Hypertension, Gestational Hypertension, Preeclampsia, and preeclampsia superimposed on chronic hypertension)".

**Chronic underlying hypertension** – "is diagnosed in women with documented blood pressure>= 140/90mm Hg before pregnancy or before 20 weeks of gestation or both".

**Gestational hypertension** -" is when the blood pressure in pregnant women reaches 140/90mm Hg or more significant for the first time after mid-pregnancy, but in whom proteinuria is not identified. Gestational hypertension is classified as transient hypertension if evidence for preeclampsia does not develop and the blood pressure returns normal by 12 weeks postpartum".

**Preeclampsia superimposed on chronic hypertension**– "is when a new onset or worsening baseline hypertension is accompanied by new onset proteinuria or other findings characteristic of preeclampsia".

**Eclampsia:** is a convulsion and/or that occurs in preeclampsia mothers to which no other cause can be attributed. The seizures are generalized and may occur during, before, and after delivery.

**Wealth index:** We computed the wealth index using a principal component analysis using proxy variables such as availability of assets (e.g., land, cattle, production of cereals and other agricultural products...etc.), housing conditions (e.g., house building materials, toilet building materials, number of rooms…etc), and access to basic services (e.g., electricity, water sources…etc.). Based on the analysis, the wealth index was categorized into five categories – poorest, poor, medium, rich, and richest.

### Ethical approval and consent to participate

We obtained ethical clearance from the ethical review committee of Wachemo University with ethical reference number 992/2015 on date 03/05/2023. Permission to conduct this study on the participants of the respective hospitals was obtained from the respective hospitals' chief executive officers (medical directors). We have also obtained informed written consent from the study participants. The privacy and confidentiality of information given by each respondent were maintained, and names were recorded.

## Results

### Socio-demographic characteristics of the study participants

A total of 433 pregnant women participated in the study. The mean age of the participants was 29.4 years (SD = ±4.6). Most participants (99.1%) were married and 80.5% were Protestant. Nearly one-third (29.6%) of the participants had secondary school or above educational level. The majority 365(84.3%) of participants identified Hadiya as their ethnic origin, 219(50.6%) were housewives and more than two-thirds (68.1%) of the participants were from urban areas (Table 1).

### Obstetric and medical characteristics of study participants

Out of the pregnant women who participated in the study, 258 (59.6%) were multiparous, and one-third (33.6%) of them were between 37 and 42 gestational weeks. Three hundred thirty (76.2%) participants have attended ANC two to four times during the current pregnancy. The mean age at their 1st pregnancy was 23.3 (±2.8) years, with minimum and maximum age being 15 and 30, respectively.

Only 8(1.8%) of the study participants had a medical history of diabetic mellitus and 15(3.5%) had a history of anemia during their pregnancy. Among the study participants, 165 (38.1%) have reported history of obstetric problems such as abortion (23.1%), stillbirth (3.5%), preterm delivery (4.6%), and placenta abruption (7. 4%). Among the participants, 15(3.5%) had a family history of PIH and 30 (6.1%) had a history of chronic hypertension.

Among the participants, 6(1.4%) had a history of smoking, of which 3 (50.0%) were current smokers, and 15(3.5%) had a history of drinking alcohol, of which 8(1.8%) were current drinkers (Table 2).

**Table 1. Socio-demographic characteristics of the study participants in Hadiya zone, Ethiopia, August 1, 2023, to January 30, 2024.**

| Variables | Frequency (%) |
|---|---|
| **Age (years)** | |
| 15–20 | 12(2.8%) |
| 21–34 | 348(80.4%) |
| >=35 | 73(16.9%) |
| **Residence** | |
| Urban | 295(68.1%) |
| Rural | 138(39.1%) |
| **Ethnicity** | |
| Hadiya | 365(84.3%) |
| Gurage | 10(2.3%) |
| Kembata | 26(6.0%) |
| Amhara | 12(2.8%) |
| Silte | 10(2.3%) |
| other | 10(2.3%) |
| **Family size** | |
| 1–2 | 92(21.2%) |
| 3–4 | 147(33.9%) |
| >5 | 19444.8%) |
| **Religion** | |
| Protestant | 345(80.5%) |
| Orthodox | 50(11.5%) |
| Muslim | 24(4.5%) |
| Catholic | 9(2.1%) |
| Other | 2(0.5%) |
| **Educational status of the mother** | |
| Illiterate (Unable to read and write) | 25(5.8%) |
| Can Read &write | 63(14.8%) |
| Primary School | 108(24.9%) |
| Secondary School | 109(25.2%) |
| Above secondary level | 128(29.6%) |
| **Educational status of the husband (n = 424)** | |
| Illiterate (Unable to read and write) | 17(3.9%) |
| Can read &write | 30(6.9%) |
| Primary School | 107(24.7%) |
| Secondary school | 101(23.3%) |
| Above Secondary | 174(40.2%) |
| **Duration of marriage(years)** | |
| <=2 | 96(22.2%) |
| 3–5 | 107(24.7%) |
| 6–10 | 155(35.8%) |
| >10 | 74(17.1%) |
| **Marital status of the mother** | |
| Single | 4(0.9%) |
| Married | 429(99.1%) |
| **Occupation of the Mother** | |
| Housewife | 219(50.6%) |

*(Continued)*

**Table 1.** (Continued)

| Variables | Frequency (%) |
|---|---|
| Gov. employee | 132(30.5%) |
| Farmer | 26(6.0%) |
| Daily labourer | 43(9.9%) |
| Other | 13(3%) |
| **Highest-earning Family member's occupation** | |
| Merchant | 152(35.1%) |
| Gov. employer | 130(30.02%) |
| Farmer | 125(28.7%) |
| NGO | 24(5.5%) |
| Daily labourer | 2(0.46%) |
| **The highest education level of the family members** | |
| Ph.D. | 6(1.4%) |
| 2nd degree | 66(15.24%) |
| 1st degree | 117(27.0%) |
| Diploma | 6(1.4%) |
| Secondary School | 103(23.8%) |
| Primary school | 108(25%) |
| Can read and write | 10(2.3%) |
| Illiterate | 17(3.9%) |
| **Wealth Index** | |
| Poorest | 82(18.9% |
| Poor | 90(90%) |
| Medium | 99(22.9%) |
| Rich | 42(9.7%) |
| Richest | 120(27.7%) |

## Prevalence of pregnancy-induced hypertension

The prevalence of pregnancy-induced hypertension among women in the four hospitals of Hadiya zone was found to be 13.4% (95% CI 10.2% – 16.9%). Out of the total of women who had pregnancy-induced hypertension, 47(10.9%) were pre-eclampsia, 6(1.4%) were eclampsia and 5(1.2%) were gestational hypertension (Fig 2).

## Factors associated with pregnancy induced hypertension

A binary logistic regression analysis was conducted with a confidence level of 95% ($\alpha = 0.05$). Variables with a p-value < 0.25 were selected as candidate variables for the final model. This model aims to determine the predictors of pregnancy-induced hypertension among women attending ANC services.

From the bivariate analysis (Table 3), variables were found to be eligible to be included in the multivariable analysis.

In the multivariable logistic regression analysis, PIH was associated with wealth index, having family history of hypertension, having family history of DM, overweight maternal BMI, and gestational age (Table 4).

The odds of having PIH was eleven times (AOR = 10.99, 95% CI (3.74–32.29) higher among women with a family history of hypertension as compared to those women without a history of hypertension. The odds of having PIH was seven times (AOR = 7.67, 95% CI 2.22–26.45) higher among women with a family history of DM as compared to those women without a history of DM. The odds of having PIH was four times (AOR = 4.41 at 95% CI 2.11–9.10) higher among

**Table 2. Obstetric and medical characteristics of women attending antenatal care at public Health Hospitals, Hadiya zone, Ethiopia, August 1, 2023, to January 30, 2024 (N = 433).**

| Variables | Frequency (%) |
|---|---|
| **Parity** | |
| Nulliparous | 57(13.2%) |
| Primiparous | 80(18.5%) |
| Multiparous | 258(59.6%) |
| Grand multiparous | 38(8.8%) |
| **Age at first pregnancy (years)** | |
| ≤20 | 95 (21.9%) |
| 21–25 | 238 (55.0%) |
| >25 | 100 (23.1%) |
| **Smoking Habit** | |
| Yes | 6(1.4%) |
| Before pregnancy | 3(0.7%) |
| Both | 3(0.7%) |
| No | 427(98.6%) |
| **Alcohol Drinking** | |
| Yes | 15(3.5%) |
| Before | 7(1.6%) |
| Both before and after | 8(1.8%) |
| No | 418(96.5%) |
| **History of placenta abruption(n = 376)** | |
| Yes | 32(7.4%) |
| No | 344(92.6%) |
| **Abortion** | |
| Yes | 100(23.1%) |
| No | 333(76.9%)98 |
| **Type of abortion (n = 100)** | |
| Spontaneous | 88(88.0%) |
| Induced | 12(12.0%) |
| **History of preterm delivery (n = 376)** | |
| Yes | 20(4.6%) |
| No | 356(82.2%) |
| **History of stillbirth (n = 376)** | |
| Yes | 13(3.5%) |
| No | 363(96.5%) |
| **Gestational age at interview** | |
| 20–28 week | 138(31.9%) |
| 29–32 week | 69 (15.9%) |
| ≥33 week | 226 (52.2%) |
| **Sign of pregnancy complication** | |
| Yes | 165(38.1%) |
| No | 268(61.9.0%) |
| **Gestational age at1st ANC** | |
| ≤20 weeks21–28 weeks | 282 (65.1%) |
| 29–32 weeks | 119 (27.5%) |
| ≥33 weeks | 25 (5.8%) |

*(Continued)*

**Table 2.** (Continued)

| Variables | Frequency (%) |
|---|---|
| | 7 (1.6%) |
| **Number of ANC visit** | |
| 1 | 65(15%) |
| 2–4 | 330(76.2%) |
| ≥5 | 38(8.8%) |
| **Change of husband before this pregnancy** | |
| Yes | 11(2.5%) |
| No | 422(97.5%) |
| **BMI** | 319(73.7%) |
| Normal | 114(26.3% |
| Overweight | |
| **Medical history of diabetes mellitus** | |
| Yes | 8(1.8%) |
| No | 425(98.2%) |
| **Anaemia** | |
| Yes | 15(3.5%) |
| No | 418(96.5%) |
| **Family history of PIH** | |
| Yes | 15 (3.5%) |
| No | 418 (96.5%) |
| **Family history of hypertension** | |
| Yes | 30(6.9%) |
| No | 403(93.1%) |
| **Family history of DM** | |
| **Yes** | 32(7.4%) |
| **No** | 401(92.6%) |

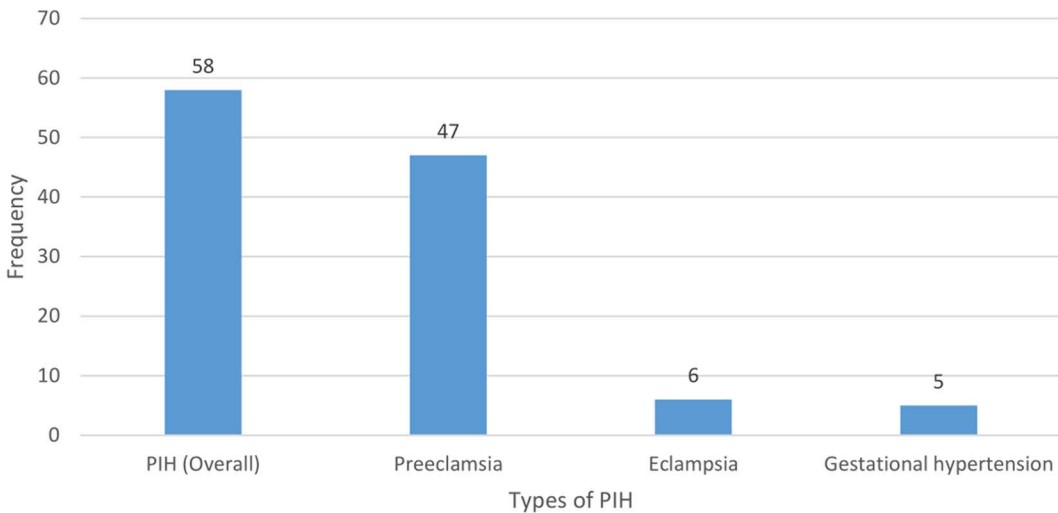

**Fig 2. Pregnancy-induced hypertension (PIH) among pregnant women in Hadiya zone, Ethiopia, August 1, 2023, to January 30, 2024.**

**Table 3. Bivariate analysis showing candidate variables for the multivariable analysis to identify factors associated with PIH among women attending antenatal care at public Health Hospitals, Hadiya zone, Ethiopia, August 1, 2023, to January 30, 2024 (N = 433).**

| Variable | PIH | | COR (95% CI) | P value |
|---|---|---|---|---|
| | Yes | No | | |
| **Age of the mother** | | | | |
| 15-20 | 0 | 12 | | |
| 21-34 | 46 | 302 | 0.78(0.39-1.547) | 0.47 |
| >=35 | 12 | 61 | 1 | |
| **Educational status of the mother** | | | | |
| Illiterate (Unable to read and write) | 8 | 17 | 1 | **0.009** |
| Can Read &write | 9 | 54 | 3.832 (1.400-10.490) | |
| Primary School | 15 | 93 | 1.352 (0.553-3.331) | |
| Secondary School | 12 | 97 | 1.313(0.603-2.860) | |
| Above secondary level | 14 | 114 | 1.007(0.445-2.860) | |
| **Occupation of the Mother** | | | | |
| Housewife | 39 | 180 | 1 | **0.027** |
| Gov. employee | 12 | 120 | 0.462(0.232-0.917) | |
| Farmer | 2 | 24 | 0.385(0.087-1.695) | |
| Student | 5 | 38 | 0.607(0.225-1.642) | |
| Other | 0 | 13 | 0.000 | |
| **Wealth Index** | | | | |
| Poorest | 20 | 62 | 1 | |
| Poor | 6 | 84 | 0.221(0.084-0.584) | **0.002** |
| Medium | 13 | 86 | 0.461(0.217-1.013) | **0.054** |
| Rich | 4 | 38 | 0.326 (0.104-1.027) | 0.56 |
| Richest | 15 | 105 | 0.443(0.211-0.928) | **0.031** |
| **Parity** | | | | |
| Nulliparous | 9 | 48 | 1.24(0.38-4.03) | 0.72 |
| Primiparous | 6 | 74 | 0.53(0.152-1.88) | 0.33 |
| Multiparous | 38 | 220 | 1.14(0.42-3.1) | 0.79 |
| Grand multiparous | 5 | 33 | 1.00 | |
| **Gestational age at interview** | | | | |
| 20 – 28 week | 7 | 131 | 3.55.310-9.617) | **0.013** |
| 29 – 32 week | 11 | 58 | 4.03(1.749-9.265) | **0.001** |
| ≥33 week | 40 | 186 | 1.00 | **1.00** |
| **BMI** | | | | |
| 18.6 -24.5 | 25 | 294 | 1.00 | **0.000** |
| 24.6-29.5 | 33 | 81 | 4.79(2.696-8.514) | |
| **Family history of hypertension** | | | | |
| Yes | 18 | 40 | 13.612(6.116-30.298) | **0.000** |
| No | 12 | 363 | 1.00 | |
| **Family history of DM** | | | | |
| **Yes** | 18 | 40 | 0.86(0.04-0.17) | **0.000** |
| **No** | 14 | 361 | 1.00 | |

**Table 4. Multivariate logistic regression analysis of factors associated with PIH among pregnant women attending antenatal care at public Health Hospitals, Hadiya zone, Ethiopia, August 1, 2023, to January 30, 2024(N = 433).**

| Variable | PIH | | COR (95% CI) | AOR (95% CI) |
|---|---|---|---|---|
| | Yes | No | | |
| **Wealth Index** | | | | |
| Poorest | 20 | 62 | 1 | 1 |
| Poor | 6 | 84 | 0.221(0.084-0.584) | **0.28(0.081-0.99)** |
| Medium | 13 | 86 | 0.461(0.217-1.013) | 1.01(0.318-3.22) |
| Rich | 4 | 38 | 0.326 (0.104-1.027) | 0.28(0.053-1.46) |
| Richest | 15 | 105 | 0.443(0.211-0.928) | 0.91(0.280-2.980) |
| **Gestational age at interview** | | | | |
| 20–28 week | 7 | 131 | 3.549(1.310-9.617) | **0.21(0.056-0.824)** |
| 29–32 week | 11 | 58 | 4.025(1.749-9.265) | 1.48(0.58-3.78) |
| ≥33 week | 40 | 186 | | **1** |
| **BMI** | | | | |
| 18.6–24.5 | 25 | 294 | | **1** |
| 24.6–29.5 | 33 | 81 | 4.791(2.696-8.514) | **4.41(2.11-9.10)** |
| **Family history of hypertension** | | | | |
| Yes | 18 | 40 | 13.612(6.116-30.298) | **10.99(3.74-32.29)** |
| No | 12 | 363 | | **1** |
| **Family history of DM** | | | | |
| Yes | 12 | 46 | 0.86(0.04-0.17) | **7.67(2.22-26.453)** |
| No | 11 | 410 | | **1** |

overweight women than those with normal weight. The odds of having PIH was 79% (AOR = 0.21 at 95% CI 0.056–0.82) higher among women whose gestational age greater than 33 weeks than women whose gestational age 29−32 weeks. The odds of having PIH was 72%(AOR = 0.28 at 95% CI 0.081–0.99) higher among women whose poorwealth index status as compared to those with poorest wealth index status.

## Discussion

The principal findings of this study revealed that more than 13 out of 100 pregnant women (13.4%) develop PIH among women attending antenatal care in public hospitals in the Hadiya zone, Central Ethiopia Regional State and this is associated with being in the poorest wealth index category, gestational age, overweight, and family history of hypertension. This prevalence of PIH is consistent with other similar studies, for example, a study in Addis Ababa found a prevalence of (11.5%) while studies in South Africa and Brazil reported 12% and 13.9%, respectively. However, it is higher than study findings from Mizan-tepi (7.5%), Tikuranbesa Hospital (5.3%) and Mettu Karl Hospital (2.4%). Furthermore, the prevalence in this study is still lower than the findings of a prospective cohort study conducted in the Tigray regional state, which reported (66.4%) and Zimbabwe which reported (19.5%). The discrepancy in the findings could be attributed to the differences in settings, study period, and socio-economic characteristics of the pregnant women. Additionally, differences in healthcare access, cultural practices, and dietary habits within different regions of Ethiopia could contribute to variations in PIH prevalence and associated factors.

Pregnancy-induced hypertension has critical public health, clinical, and societal implications. PIH is one of the most common causes of maternal mortality. Death of a woman at her productive, childbearing, and rising age can severely affect the well-being of society. Maternal mortality has also negative repercussions on the community's reliance on the health system as it may create misunderstandings and myths in the community which can affect the health-seeking

behaviour of the women. Thus, the local health authorities, clinicians, and public health practitioners must work towards early detection and treatment of PIH, create awareness on PIH among the general population, and strive to enhance the quality of antenatal care services in healthcare facilities.

In our study, the odds of developing PIH were associated with a pregnant woman being in the poorest wealth index category. The association of low socioeconomic status and inadequate access to healthcare services are reported to be critical contributors to higher rates of PIH in other studies conducted in Ethiopia [20], Bangalore [21], Pakistan [22], Sweden [23]. Women with lower socioeconomic status might have lower levels of health literacy (education), lower levels of health-seeking practices, and higher levels of stress which may increase the probability of developing hypertension (PIH). Targeting women with lower socioeconomic status to enhance their awareness of the need to attend antenatal care and early screening of their status is imperative.

Consistent with other studies [15,19,24],this study reported that overweight/obesity is associated with PIH. Overweight/obesity is one of the known risk factors of hypertension as it causes stenosis in the blood vessels – causing resistance to the blood flow and increasing blood pressure. In pregnant women, the blood volume increases by about 45% above the non-pregnant level [25]. This expansion in blood volume during pregnancy coupled with stenosed blood vessels among those with overweight/obesity will result in the development of PIH. Women must consider maintaining optimal weight before they conceive and follow appropriate measures such as dietary modifications and recommended exercise to control their weight gains during pregnancy.

Pregnant women with a family history of diabetes mellitus were about seven times as likely to develop PIH. The findings in the present study align with research conducted in Ethiopia [26], USA [27], and Thailand [28]. Genetic factors may predispose women to an increased risk of PIH.

A family history of hypertension and PIH is known to be associated with PIH. As indicated by several reviewed articles [6,19,29], hypertension is known to be a hereditary condition that has a genetic component. Pregnant women with a familial history of hypertension or PIH need to be aware of the increased risks and implement hypertension preventive measures such as avoiding smoking and alcohol, dietary modifications, and physical activity.

This study is one of the few conducted in the area and includes all the public hospitals located in the zone. Its findings may represent a broader geographical area, encompassing both urban and rural regions, unlike other studies that were conducted in a single public hospital [11,30].

However, this study might be limited due to the potential introduction of recall bias during the interview and social desirability bias, as women might have intended to respond positively to the interview questions. Moreover, as we have included participants in the study consecutively, there might have been selection bias that might have limited the generalizability of this study. Efforts were made to mitigate these limitations by using specific and targeted questions.

## Conclusions

More than 13 out of 100 pregnant women were found to have pregnancy-induced hypertension in the Hadiya zone, Southern Ethiopia. The high prevalence of PIH and its association with wealth index, gestational age, overweight/obesity and family history of hypertension/DM underscores the need for targeted, tailored, and contextualized interventions to address PIH in the Hadiya zone. Creating awareness of PIH among pregnant women and the general population and enhancing early screening and detection services in the community and healthcare facilities need to be strengthened in the Hadiya zone.

## Supporting information

**S1 File.**
(DOCX)

## Acknowledgments

Special thanks are extended to the Central University of Tamil Nadu (CUTN), India, and the Indian Council for Cultural Relations (ICCR) scholarship program in collaboration with Wachemo University. We would also like to thank the management and staff of Nigist Elleni Mohammed Memorial Comprehensive Specialized Hospital (NEMMCSH), Shone Primary Hospital, Gimbichu Primary Hospital, and Homecho Primary Hospital for their frequent consultations during my thesis work. Additionally, we extend our gratitude to all the data collectors, supervisors, and study participants who contributed to this research.

## Author contributions

**Conceptualization:** Zerfework Debebe Argago, Nebiyu Dereje, Neena Elezebeth Philip.

**Data curation:** Zerfework Debebe Argago, Neena Elezebeth Philip.

**Formal analysis:** Zerfework Debebe Argago.

**Supervision:** Nebiyu Dereje, Neena Elezebeth Philip.

**Visualization:** Nebiyu Dereje.

**Writing – original draft:** Zerfework Debebe Argago.

**Writing – review & editing:** Nebiyu Dereje, Neena Elezebeth Philip.

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
