## [Decision Letter · Decision Letter 0]

5 Nov 2024

PONE-D-24-43110Prevalence and Associated Factors of Pregnancy Induced Hypertension among Pregnant Women in Public Hospitals of Hadiya zone, Southern Ethiopia: A cross-sectional studyPLOS ONE

Dear Dr.Debebe Argago

Thank you for submitting your manuscript to PLOS ONE. After careful consideration, we feel that it has merit but does not fully meet PLOS ONE’s publication criteria as it currently stands. Therefore, we invite you to submit a revised version of the manuscript that addresses the points raised during the review process.

We look forward to receiving your revised manuscript.

Kind regards,

Adera Debella Kebede, MSC

Academic Editor

PLOS ONE

Journal Requirements:

Reviewers' comments:

Reviewer's Responses to Questions

**Comments to the Author**

1. Is the manuscript technically sound, and do the data support the conclusions?

Reviewer #1: Yes

Reviewer #2: Yes

2. Has the statistical analysis been performed appropriately and rigorously? 

Reviewer #1: Yes

Reviewer #2: Yes

3. Have the authors made all data underlying the findings in their manuscript fully available?

Reviewer #1: Yes

Reviewer #2: Yes

4. Is the manuscript presented in an intelligible fashion and written in standard English?

Reviewer #1: No

Reviewer #2: No

5. Review Comments to the Author

Reviewer #1: Dear Authors,

Thank you for your valuable contribution to the literature on pregnancy-induced hypertension (PIH). Your study, "Prevalence and Associated Factors of Pregnancy Induced Hypertension among Pregnant Women in Public Hospitals of Hadiya zone, Southern Ethiopia: A cross-sectional study," addresses a critical public health issue and provides important insights into the risk factors for PIH in this region.

Your findings highlight the significant burden of PIH in Hadiya zone and underscore the need for targeted interventions to improve maternal health outcomes. The identification of key risk factors, such as wealth index, gestational age, overweight/obesity and family history of hypertension, can help healthcare providers to identify high-risk women and implement timely preventive measures.

While your study has made a valuable contribution, I would like to offer a few suggestions for improvement:

- A Systematic Review and Meta-Analysis on PIH in Ethiopia Already Exists. This is a valid concern. While your study provides valuable insights into the local context of Hadiya zone, it's crucial to differentiate your contribution from existing systematic reviews and meta-analyses. By focusing on these aspects, you can add value to the existing body of knowledge and provide actionable recommendations for improving maternal health outcomes. So, authors should be incorporate these suggestion in back ground of the study to strengthen your study.

- In study setting and design – “On average, the total number of mothers serviced by all hospitals in 2011 was 10,774. The data provided is outdated (2011) and may not reflect the current situation. It is recommended to use more recent data for accurate information.

- Source and study population

“The study population consisted of consented pregnant women who visited ANC follow-up in the selected hospitals of the Hadiya zone, and whose gestational age was greater than 20 weeks during the study period”. It’s confused me. What mean the phrase “consisted of consented pregnant women?”

- Mothers who have a previous history of hypertension were excluded from the study. Why? do you have justification

- Sampling procedure section—last sentences “pregnant women attending ANC in the respective hospital were included consecutively until the desired sample size was reached’. The authors chose to use consecutive sampling, which involves including all eligible participants who meet the inclusion criteria until the desired sample size is reached. Why not sampled? And also, not acknowledge the potential for selection bias and limited generalizability in the discussion section.

- Data collection tools and procedures

o The sentences... Data on proteinuria and other clinical variables was taken from the medical charts of pregnant women. However, is proteinuria data, a critical factor in classifying different types of pregnancy-induced hypertension (PIH), readily accessible from the medical charts of all pregnant women? This information allowed for accurate categorization of PIH cases based on established diagnostic criteria. so, Why was not taken a sample directly from women?

o Measurement of blood pressure was taken by using standard procedures, with a digital sphygmomanometer. Better if more elaborate the standard procedure for measuring blood pressure using a digital sphygmomanometer for the general audience easily understanding

- Data Processing and Analysis

o Those variables with a P value ≤0.25 in the bivariable analysis... what is your baseline reference to take a P value ≤0.25? Why not take 0.2, 0.3, 0.4, etc.? Do you have any justification?

- Study variable

o Obstertric variable—the variable “mode of delivery’- it is the previous or current mode of delivery? Additionally, I did not see any figures about this variable in result sections.

- Operational definitions

o There was no real definition of PIH used. “Pregnancy-induced hypertension (PIH) "is systolic blood pressure ≥ 140 mm Hg or diastolic blood pressure ≥ 90 mmHg after gestation age of twenty weeks in a previously average hypertensive woman". This definition contradicts the definition written in 1st paragraph of your background, “Pregnancy-induced hypertension (PIH) "is systolic blood pressure ≥ 140 mm Hg or diastolic blood pressure ≥ 90 mmHg after gestation age of twenty weeks in a previously normal BP". It would be important to define clearly and should be referenced where it take the definition.

o Better if define wealth index

o Elaborate how to measure the BMI

o Author’s defined HELLP Syndrome: It is a variant of preeclampsia characterized by hemolysis (H), elevated liver enzymes (EL), and low platelet count (LP). But I did see any where those are written in manuscript. So, what importance this variable, better if removed.

Result

- Grammar error….Most participants (429, 99.1%) were married. Edit as Most participants (99.1%) were married

- In multivariate of factors associated with PIH; the CI of variable family of history DM is wider (7.67(2.22-26.453)). Why? do you have justification

Discussion

- Authors compared the finding of prevalence of PIH within Ethiopia only. Better if additionally compared with Sub-Saharan Africa, globally.

- Again authors said… The discrepancy in the findings could be attributed to the differences in settings, study period, and socio-economic characteristics of the pregnant women. While the all studies were conducted in Ethiopia, there could be variations in specific methodologies, sample sizes, and data collection techniques that might influence the findings rather than study setting. Additionally, subtle differences in healthcare access, cultural practices, and dietary habits within different regions of Ethiopia could contribute to variations in PIH prevalence and associated factors. It's important to critically evaluate the methods and limitations of each study to identify potential sources of discrepancy. By carefully considering these factors, researchers can better understand the nuances of PIH in different Ethiopian contexts and develop more targeted prevention and management strategies.

- Discussion section para 2 seems to be the conclusion. I recommend the authors combine the second paragraph of the discussion section with the conclusion section to improve the flow and clarity of the manuscript

Reviewer #2: 1. First sentence of the background should be referenced

2. Please critically evaluate all of the document for line spaces and English grammar

3. There should be line number in the document to easy the review process

4. In the method part you state that your study area is Hadiya Zone of the Central Ethiopia Regional State in south-western Ethiopia, but on your title it is southern Ethiopia. Please be consistent throughout the manuscript

5. Why you used small a margin of error of 3%? Because a smaller margin of error indicates more confidence in the accuracy of the results.

6. You mentioned that data on proteinuria and other clinical variables was taken from the medical charts of pregnant women. What do you do if the data on proteinuria did not available in medical charts? Is proteinuria important for the diagnosis of PIH?

7. Why dependent variable (Pregnancy-Induced hypertension) is yes/no (1=Yes,0=No)?. Why not be 1=BP>140mmhg/90mmhg, 0=<140mmhg/90mmhg?

8. Important variables like wealth index is not operationalized. Or how wealth index was computed?

9. Measurement of your dependent variable is not clearly mentioned in method part. You have to add measurement sub-title in your method part.

10. You only stated that blood pressure was measured by a digital Sphygmomanometer on two measurements among pregnant women with gestational age >20 weeks in abstract part. But you did not clearly state what time gap there b/n two measurements. This and other should be explicitly described in measurement part.

11. Figures and tables did not labelled correctly. Please try to organize the tables and figures with their respective titles

12. Figure 1 has not been referenced in the main text of your manuscript file. If tables and figures are not cited in the manuscript they will not appear during publication

13. Titles of figures and tables listed in page number 22 should be written near the respective figures and tables.

14. Discussion part is good but needs some improvement with reasons and evidence

15. Please include URL for Some references (e.g. ref 2 and 6)

6. PLOS authors have the option to publish the peer review history of their article (what does this mean? ). If published, this will include your full peer review and any attached files.

**Do you want your identity to be public for this peer review?** For information about this choice, including consent withdrawal, please see our Privacy Policy .

Reviewer #1: **Yes: ** Ebisa Zerihun

Reviewer #2: No

---

## [Author Response · Author response to Decision Letter 1]

16 Dec 2024

Point-to-point responses

Dear Adera Debella Kebede, MSC

Academic Editor

PLOS ONE

Thank you for the opportunity to review our manuscript. We thank you and the reviewers for the insightful comments that have improved our manuscript. We have addressed all the comments and submitted the revised version of the manuscript to the online portal. Please kindly receive our point-by-point responses to the comments as follows:

Journal Requirements:

Response: We have formatted now as per the PLOS ONE's style requirements.

2. Please include a complete copy of PLOS’ questionnaire on inclusivity in global research in your revised manuscript. And submit it as a supporting file.

Response: We have completed the PLOS’ questionnaire on inclusivity in global research and submitted it as a supporting file.

Response: This is well-noted, and we will share our dataset immediately following the acceptance of our manuscript.

Response: We have included the ethics section only in the methods section.

Response to Reviewer 1 comments

1. Thank you for your valuable contribution to the literature on pregnancy-induced hypertension (PIH). Your study, "Prevalence and Associated Factors of Pregnancy Induced Hypertension among Pregnant Women in Public Hospitals of Hadiya zone, Southern Ethiopia: A cross-sectional study," addresses a critical public health issue and provides important insights into the risk factors for PIH in this region.

Your findings highlight the significant burden of PIH in Hadiya zone and underscore the need for targeted interventions to improve maternal health outcomes. The identification of key risk factors, such as wealth index, gestational age, overweight/obesity and family history of hypertension, can help healthcare providers to identify high-risk women and implement timely preventive measures.

Response: Thank you for your positive reflections.

2. A Systematic Review and Meta-Analysis on PIH in Ethiopia Already Exists. This is a valid concern. While your study provides valuable insights into the local context of Hadiya zone, it's crucial to differentiate your contribution from existing systematic reviews and meta-analyses. By focusing on these aspects, you can add value to the existing body of knowledge and provide actionable recommendations for improving maternal health outcomes. So, authors should be incorporate these suggestions in back ground of the study to strengthen your study.

Response: Thank you for the comment. We now added a justification to substantiate the need to conduct this study in the Hadiya zone. We added the following sentence in the introduction:

As Ethiopia is diverse in terms of cultural practices, socio-economic status (e.g., wealth index), and access to healthcare facilities, it is critical to understand contextual factors that affect PIH.

3. In study setting and design – “On average, the total number of mothers serviced by all

hospitals in 2011 was 10,774. The data provided is outdated (2011) and may not reflect the current situation. It is recommended to use more recent data for accurate information.

Response: This was by mistake. It was to mean 2021. We now have revised it.

4. Source and study population “The study population consisted of consented pregnant women who visited ANC follow-up in the selected hospitals of the Hadiya zone, and whose gestational age was greater than 20 weeks during the study period”. It’s confused me. What mean the phrase “consisted of consented pregnant women?”

Response: Thank you. We have revised the sentence now for clarity.

5. Mothers who have a previous history of hypertension were excluded from the study. Why? do you have justification?

Response: Thank you. The definition of PIH considers hypertension that occurs after 20 weeks of gestation in women with previously normal blood pressure.

6. Sampling procedure section—last sentences “pregnant women attending ANC in the respective hospital were included consecutively until the desired sample size was reached’. The authors chose to use consecutive sampling, which involves including all eligible participants who meet the inclusion criteria until the desired sample size is reached. Why not sampled? And also, not acknowledge the potential for selection bias and limited generalizability in the discussion section.

Response: Thank you for picking up this point. As we have looked at a limited period, the consecutive sampling method was ideal for us. However, we note the limitations of this method in terms of generalizability and potential selection bias. We have added a sentence in the limitations of the study as follows:

Moreover, as we have included participants in the study consecutively, there might have been selection bias that might have limited the generalizability of this study.

7. Data collection tools and proceduresThe sentences... Data on proteinuria and other clinical variables was taken from the medical charts of pregnant women. However, is proteinuria data, a critical factor in classifying different types of pregnancy-induced hypertension (PIH), readily accessible from the medical charts of all pregnant women? This information allowed for accurate categorization of PIH cases based on established diagnostic criteria. so, why was not taken a sample directly from women?

Response: Thank you. Urine analysis, including proteinuria, is a standard and routine investigation for all women attending ANC and is readily available from the charts. The repetitive tests are not recommended for ethical, cost, and safety implications.

8. Measurement of blood pressure was taken by using standard procedures, with a digital Sphygmomanometer. Better if more elaborate the standard procedure for measuring blood Pressure using a digital sphygmomanometer for the general audience easily understanding

Response: Thank you. We now have included details on the BP measurement procedures as follows:

The woman was instructed to sit quietly and rest for 15 minutes with their legs uncrossed. The measurements were taken with the participant sitting upright with proper back support and with the right arm supported on a table or a surface at the level of the heart. The cuff was inflated at 2–3 mmHg per second. Systolic blood pressure (SBP) was taken up on hearing the first sound, and diastolic blood pressure (DBP) was taken up on the fourth (muffled) Korotkoff sound. Three readings were taken at five-minute intervals, and the average was used as the final blood pressure measurement.

9. Data Processing and Analysis

Those variables with a P value ≤0.25 in the bivariable analysis... what is your baseline reference to take a P value ≤0.25? Why not take 0.2, 0.3, 0.4, etc.? Do you have any justification?

Response: It is a standard procedure in building models to include a cut-off value of ≤0.25 for p-values for individual variables in the bivariate analysis. The reason for this is not to exclude variables that may be of significance too early in their analysis. The criterion of a p-value ≤ 0.25 is less stringent than the conventional p-value of 0.05. This approach enables the inclusion of variables that may not demonstrate statistical significance in bivariate analysis but could show significance when examined in combination with other variables in a multivariable model. Adopting a higher threshold (such as 0.25) minimizes the risk of omitting essential variables that, while having borderline significance, remain relevant within the context of the entire model. According to Hosmer and Lemeshow’s (2000) in their book Applied Logistic Regression, a p-value of ≤ 0.25 is recommended to prevent the premature exclusion of variables, ensuring that potential predictors are not overlooked too early in the analysis. P value = 0.20 or lower could risk excluding relevant variables too soon and P value = 0.30 or higher could include too many irrelevant variables, which can lead to overfitting or decreased representation productivity. As a result, a p-value of ≤ 0.25 effectively balances the inclusion of potentially essential variables while filtering out those unlikely to be helpful in multivariate analysis.

10. Study variable

Obstetrics variable—the variable “mode of delivery’- it is the previous or current mode of delivery? Additionally, I did not see any figures about this variable in result sections.

Response: Our questionnaire obtained comprehensive information concerning the mode of delivery in the past. However, pregnant women who fall under the category of nullipara (women who have never delivered) and primigravida (first pregnancy) were omitted as they had no relevant histories regarding delivery. Such a restriction is essential in making sure that the data we analyze is representative of the participants and does not compromise the validity of our conclusions about the specific groups.

11. There was no real definition of PIH used. “Pregnancy-induced hypertension (PIH) "is systolic blood pressure ≥ 140 mm Hg or diastolic blood pressure ≥ 90 mmHg after gestation age of twenty weeks in a previously average hypertensive woman". This definition contradicts the definition written in 1st paragraph of your background, “Pregnancy-induced hypertension (PIH) "is systolic blood pressure ≥ 140 mm Hg or diastolic blood pressure ≥ 90 mmHg after gestation age of twenty weeks in a previously normal BP". It would be important to define clearly and should be referenced where it take the definition.

Response: Thank you. We make the necessary corrections in both the introductory paragraph of the background section and in the operational definition according to ACOG."

"Pregnancy-induced hypertension (PIH) is defined by the American College of Obstetricians and Gynecologists (ACOG) as hypertension that occurs after 20 weeks of gestation in women who previously had normal blood pressure. Specifically, hypertension is characterized by a systolic blood pressure of 140 mm Hg or higher, a diastolic blood pressure of 90 mm Hg or higher, or both, measured on two separate occasions at least four hours apart".

12. Better if define wealth index

Response: Thank you. We now have defined the wealth index as follows:

We computed the wealth index using a principal component analysis using proxy variables such as availability of assets (e.g., land, cattle, production of cereals and other agricultural products...etc.), housing conditions (e.g., house building materials, toilet building materials, number of rooms…etc), and access to basic services (e.g., electricity, water sources…etc.). Based on the analysis, the wealth index was categorized into five categories - poorest, poor, medium, rich, and richest.

13. Elaborate how to measure the BMI

Response: Thank you. We have included it now.

14. Author’s defined HELLP Syndrome: It is a variant of preeclampsia characterized by hemolysis (H), elevated liver enzymes (EL), and low platelet count (LP). But I did see any where those are written in manuscript. So, what importance this variable, better if removed.

Response: Thank you. We have removed it now.

15. Result

- Grammar error….Most participants (429, 99.1%) was married. Edit as most participants (99.1%) were married

Response: Thank you. We have corrected it now.

16. In multivariate of factors associated with PIH; the CI of variable family of history DM is wider (7.67(2.22-26.453)). Why? do you have justification

Response: The wider confidence interval (CI) for the variable "family history of diabetes mellitus (DM) associated with" pregnancy-induced hypertension (PIH), noted as 7.67 (2.22-26.453), can be justified by several factors. Firstly, a wide CI often indicates more significant variability in the data, suggesting that multiple confounding factors may affect the influence of a family history of DM on PIH. Secondly, the sample size is small for participants with a family history of DM, it can lead to less reliable estimates and, thus, a wider CI. Additionally, lower statistical power due to fewer cases may contribute to this width and the complexity of the relationship between the family history of DM and PIH, which could involve genetic and environmental influences. Overall, the wide CI reflects uncertainty in the association and highlights the need for further investigation to clarify its significance.

17. Discussion: Authors compared the finding of prevalence of PIH within Ethiopia only. Better if additionally compared with Sub-Saharan Africa, globally

Response: Thank you. We have included comparisons of our findings with other settings from SSA and globally. However, we need to make sure that we are comparing our findings with a setting that is compatible with our setting.

18. Again authors said… The discrepancy in the findings could be attributed to the differences in settings, study period, and socio-economic characteristics of the pregnant women. While the all studies were conducted in Ethiopia, there could be variations in specific methodologies, sample sizes, and data collection techniques that might influence the findings rather than study setting. Additionally, subtle differences in healthcare access, cultural practices, and dietary habits within different regions of Ethiopia could contribute to variations in PIH prevalence and associated factors. It's important to critically evaluate the methods and limitations of each study to identify potential sources of discrepancy. By carefully considering these factors, researchers can better understand the nuances of PIH in different Ethiopian contexts and develop more targeted prevention and management strategies

Response: Thank you. This is well noted, and revisions are made accordingly.

19. Discussion section para 2 seems to be the conclusion. I recommend the authors combine the second paragraph of the discussion section with the conclusion section to improve the flow and clarity of the manuscript

Response: Thank you. The second paragraph of the discussion section speaks about the overall implications of the findings.

Response to Reviewer #2

1. First sentence of the background should be referenced

Response: Thank you. We now have referenced it.

2. Please critically evaluate all of the document for line spaces and English grammar

Response: Thank you. We have revised it for appropriate formatting and grammar.

3. There should be line number in the document to easy the review process

Response: Okay, we have included line numbers now.

4. In the method part you state that your study area is Hadiya Zone of the Central Ethiopia Regional State in south-western Ethiopia, but on your title it is southern Ethiopia. Please be consistent throughout the manuscript

Response: Thank you. We have corrected it now.

5. Why you used small a margin of error of 3%? Because a smaller margin of error indicates more confidence in the a

---

## [Decision Letter · Decision Letter 1]

24 Apr 2025

PONE-D-24-43110R1Prevalence and Associated Factors of Pregnancy Induced Hypertension among Pregnant Women in Public Hospitals of Hadiya Zone, Central Ethiopia: A Cross-Sectional StudyPLOS ONE

Dear Dr. Debebe Argago,

Thank you for submitting your manuscript to PLOS ONE. After careful consideration, we feel that it has merit but does not fully meet PLOS ONE’s publication criteria as it currently stands. Therefore, we invite you to submit a revised version of the manuscript that addresses the points raised during the review process.

We look forward to receiving your revised manuscript.

Kind regards,

Jianhong Zhou

Staff Editor

PLOS ONE

Journal Requirements:

Reviewers' comments:

Reviewer's Responses to Questions

**Comments to the Author**

1. If the authors have adequately addressed your comments raised in a previous round of review and you feel that this manuscript is now acceptable for publication, you may indicate that here to bypass the “Comments to the Author” section, enter your conflict of interest statement in the “Confidential to Editor” section, and submit your "Accept" recommendation.

Reviewer #1: All comments have been addressed

Reviewer #2: All comments have been addressed

2. Is the manuscript technically sound, and do the data support the conclusions?

Reviewer #1: Yes

Reviewer #2: Yes

3. Has the statistical analysis been performed appropriately and rigorously? 

Reviewer #1: Yes

Reviewer #2: Yes

4. Have the authors made all data underlying the findings in their manuscript fully available?

Reviewer #1: Yes

Reviewer #2: Yes

5. Is the manuscript presented in an intelligible fashion and written in standard English?

Reviewer #1: No

Reviewer #2: (No Response)

6. Review Comments to the Author

Reviewer #1: Dear authors, Thanks for addressing all my comments. I recommend you 1. Please ensure that your manuscript meets PLOS ONE's style requirements. 2. Please move your ethical approval statement to the end of methods sections. 3. Authors should correct in manuscript some typographical and grammatical errors before publication.

Reviewer #2: All comments have been addressed. except, URL is not added in reference numbers 3 and 7 in the revised manuscript.

7. PLOS authors have the option to publish the peer review history of their article (what does this mean? ). If published, this will include your full peer review and any attached files.

**Do you want your identity to be public for this peer review?** For information about this choice, including consent withdrawal, please see our Privacy Policy .

Reviewer #1: **Yes: ** Ebisa Zerihun

Reviewer #2: No

---

## [Author Response · Author response to Decision Letter 2]

28 Apr 2025

Point-to-point responses

Dear Adera Debella Kebede, MSC

Academic Editor

PLOS ONE

Thank you for the opportunity to review our manuscript. We thank you and the reviewers for the insightful comments that have improved our manuscript. We have addressed all the comments and submitted the revised version of the manuscript to the online portal. Please kindly receive our point-by-point responses to the comments as follows:

6. Review Comments to the Author

Reviewer #1: Dear authors, Thanks for addressing all my comments. I recommend you 1. Please ensure that your manuscript meets PLOS ONE's style requirements. 2. Please move your ethical approval statement to the end of methods sections. 3. Authors should correct in manuscript some typographical and grammatical errors before publication.

Response to Reviewer 1 comments

1. Please ensure that your manuscript meets PLOS ONE's style requirements.

Response: We have formatted as per the PLOS ONE's style requirements.

2. Please move your ethical approval statement to the end of methods sections.

Response: We moved the ethical approval statement to the end of the methods

section.

3.Authors should correct in manuscript some typographical and grammatical errors before

publication.

Response: We corrected some typographical and grammatical errors in the

manuscript.

Reviewer #2: All comments have been addressed. except, URL is not added in reference numbers 3 and 7 in the revised manuscript.

Response to Reviewer 2 comments

Response: We have added the URLs for reference numbers 3 and 7 in the revised

manuscript.

7. PLOS authors have the option to publish the peer review history of their article (what does this mean?). If published, this will include your full peer review and any attached files.

Do you want your identity to be public for this peer review? For information about this choice, including consent withdrawal, please see our Privacy Policy.

Reviewer #1: Yes: Ebisa Zerihun

Reviewer #2: No

Response: No

---

## [Editor Report · Decision Letter 2]

28 May 2025

Prevalence and Associated Factors of Pregnancy Induced Hypertension among Pregnant Women in Public Hospitals of Hadiya Zone, Central Ethiopia: A Cross-Sectional Study

PONE-D-24-43110R2

Dear Dr. Debebe Argago,

We’re pleased to inform you that your manuscript has been judged scientifically suitable for publication and will be formally accepted for publication once it meets all outstanding technical requirements.

Kind regards,

Jianhong Zhou

Staff Editor

PLOS ONE
---

## [Editor Report · Acceptance letter]

PONE-D-24-43110R2

PLOS ONE

Dear Dr. Debebe Argago,

I'm pleased to inform you that your manuscript has been deemed suitable for publication in PLOS ONE. Congratulations! Your manuscript is now being handed over to our production team.

Kind regards,

on behalf of

Dr. Jianhong Zhou

Staff Editor

PLOS ONE